# Experiences of persons with Multiple Sclerosis with lifestyle adjustment–A qualitative interview study

Saskia Elkhalii-Wilhelm [1☯*], Anna Sippel [1☯], Karin Riemann-Lorenz [1], Christopher Kofahl [2], Jutta Scheiderbauer [3], Sigrid Arnade [4], Ingo Kleiter [5], Stephan Schmidt [6], Christoph Heesen [1,7]

1 Institute of Neuroimmunology and Multiple Sclerosis (INIMS), University Medical Center Hamburg-Eppendorf (UKE), Hamburg, Germany, 2 Institute of Medical Sociology, University Medical Center Hamburg-Eppendorf (UKE), Hamburg, Germany, 3 Patient Representative, Trier, Germany, 4 LEBENSNERV–Stiftung zur Förderung der Psychosomatischen MS-Forschung, Berlin, Germany, 5 Marianne-Strauß-Klinik, Behandlungszentrum Kempfenhausen für Multiple Sklerose Kranke GmbH, Berg, Germany, 6 Neurologische Gemeinschaftspraxis, Gesundheitszentrum St. Johannes, Bonn, Germany, 7 Department of Neurology, University Medical Center Hamburg-Eppendorf (UKE), Hamburg, Germany

☯ These authors contributed equally to this work.
* saskia.elkhaliiwilhelm@gmail.com

**Data Availability Statement:** The qualitative data generated and analysed during the current study are not publicly available due to privacy issues. Although pseudonyms were used to protect the

## Abstract

### Background

Persons with Multiple Sclerosis (pwMS) follow individual strategies to cope with this highly heterogeneous disease. As surveys show, lifestyle habits play an important role in pwMS. However, little is known about individual experiences of pwMS with different lifestyle adjustment strategies.

### Objective

This study aims to describe and understand individual experiences of pwMS with lifestyle adjustments.

### Methods

Semi-structured interviews were conducted with 50 pwMS in Germany. Criteria for inclusion were age ≥ 18 years and a diagnosis of relapsing-remitting Multiple Sclerosis. Data were analyzed inductively and deductively according to a six-step thematic analysis.

### Results

The three main themes for experience-based lifestyle adjustments were: 1) nutrition and supplements, 2) exercise and physical activity, and 3) stress management. Influencing factors on the decision-making process such as active disease management, information and advice, desire for mental health and social support, and the wish for self-determination were identified. Impacts of starting or maintaining lifestyle habits included, for example, MS-

participant's identity, there is still a possibility that the potential sensitive information could be identified by the details of their experiences. With the S2 Appendix all study findings can be replicated. Quantitative data (demographic) are available on reasonable request at: multiplesklerose@uke.de.

**Funding:** This study was sponsored by Roche Pharma AG. The funders had no role in study design, data collection and analysis, decision to publish, or preparation of the manuscript.

**Competing interests:** AS has received funding from Roche and salaries for talks from Novartis. CH has received research grants, congress travel compensations, and salaries for talks from Biogen, Genzyme, Sanofi-Aventis, Bayer Healthcare, Merck, Teva, Roche, and Novartis. IK has received speaker honoraria and travel funding from Bayer, Biogen, Novartis, Merck, Sanofi Genzyme, Roche; speaker honoraria from Mylan; travel funding from the Guthy-Jackson Charitable Foundation; consulted for Alexion, Bayer, Biogen, Celgene, Chugai, IQVIA, Novartis, Merck, Roche; and received research support from Chugai, Diamed. SEW, KRL, CK, JS, SA, and SS declare having no competing interests. This commercial funding does not alter our adherence to PLOS ONE policies on sharing data and materials.

specific, general, and mental health benefits, the development of coping strategies, social support, and barriers that led to a termination of lifestyle adjustments.

## Conclusion

This study provides a rich and nuanced amount of experiences of pwMS with lifestyle adjustments and leads to three important conclusions: 1) Further research is warranted to better describe the perceived effects of lifestyle habits on MS symptoms and progression, in particular with regard to nutrition and stress reduction; 2) patient education in MS should include the available evidence on lifestyle management and 3) patients need to be actively supported in changing their lifestyle behavior.

## Introduction

Multiple Sclerosis (MS) is an inflammatory and degenerative disease of the central nervous system, common in young adults between 20 and 40 years [1]. At the beginning, 85% of persons with Multiple Sclerosis (pwMS) develop episodes with neurological disabilities and full or partial recoveries, known as relapsing-remitting MS (RRMS). After 15–20 years 60 to 70% of pwMS with RRMS have converted to the secondary-progressive course (SPMS). 15% are diagnosed with primary-progressive MS (PPMS), characterized by a slow, steady increase of disability [2].

For RRMS, a wide range of disease-modifying therapies (DMTs) is available. In addition, an individual, multimodal therapeutic approach is recommended for brain health and symptom control, which includes–next to drugs and rehabilitation–also lifestyle adjustment [3].

Lifestyle factors may contribute to a large number of chronic diseases [4]. For MS, in addition to genetic factors, lifestyle factors such as smoking [5], low vitamin D levels and obesity may contribute to the disease risk [6, 7]. While the influence of lifestyle habits on MS progression in general still needs to be proven [8], low Vitamin D levels are the only factors with some weak to moderate evidence to be associated with progression [8].

Similarly, systematic reviews on the effects of specific diets and supplements on disease risk and progression in pwMS have yielded conflicting results with no conclusive evidence [9, 10]. Of note, theoretical considerations on how dietary factors could affect the highly variable progression of MS point to an approach that focusses on the influence of the dietary pattern, rather than single nutrients [11]. Finally, pwMS and researchers are notoriously interested in dietary and nutritional approaches in the treatment of MS as documented by several surveys on this topic [12–14].

Physical exercise has been investigated in detail, but most interventional studies included only small numbers of patients and investigated short-term effects. While improvement in health-related quality of life (QOL), fatigue, mood and numerous MS symptoms, such as stability of gait and prevention of falls, have been demonstrated in several systematic reviews [15–17], disease modifying or neuroprotective effects of physical exercise is still a matter of debate [8, 18] and the evidence is weak. Evidence-based guidelines for physical activity for pwMS have already been published in 2013 recommending moderate-intensity aerobic endurance as well as strength exercises for major muscle groups [19].

Since the early description of MS, stressful life events have been discussed as potential triggers of MS relapses [19], and one randomized controlled trial examining stress management has demonstrated decreased inflammatory disease activity as shown by magnetic resonance

imaging (MRI) [20]. Stress management techniques, e.g. allowing positive emotion [21], feeling self-effective and self-reliant [22, 23], may help pwMS to cope with everyday and disease-related stress. Additionally, specific relaxation methods such as progressive muscle relaxation and breathing techniques [24], meditation [25], and cognitive-behavioral stress therapy [26] may affect well-being, QOL, coping strategies and perceived stress. However, systematic reviews yield to the conclusion that evidence of the influence of stress (management) on MS remains scarce due to major qualitative differences in methodology of existing literature [27, 28]. Taken together there is some weak to moderate evidence that especially major stressful life events trigger inflammatory activity in MS.

In summary, lifestyle adjustments affect MS symptoms and possibly also the clinical course of the disease. Therefore, lifestyle adjustments should be integrated into a multimodal therapeutic concept for MS [3]. However, pwMS wishing to implement lifestyle adjustments face several difficulties: First, specific information or recommendations for lifestyle adjustments remain scarce and advice from healthcare professionals is lacking [5, 29–31]. Second, MS symptoms and other barriers may hamper lifestyle adjustments [32, 33]. A few studies of patients' experiences with lifestyle habits have rather focused on perspectives about general lifestyle risk factors [34], experiences with challenges in physical activity [35] or nutrition [36] after MS diagnosis. However, deeper insight into individual patient experiences with regard to the variety of lifestyle behaviors, the implementation of new and the maintenance of previous specific habits, is under-researched. Therefore, this study aimed to explore: 1) experiences of pwMS with different lifestyle adjustments, 2) the decision-making process concerning the type and implementation of lifestyle habits, and 3) the impact on everyday life. These findings may help to develop further approaches in patient-oriented research and for the implementation of measures supporting lifestyle adjustments.

## Materials and methods

### Ethics

The ethical committee of the Hamburg Chamber of Physicians granted ethical approval for this study (PV5770). Written consent of pwMS was obtained by neurologists or trained scientific personal.

### Participants and recruitment

This qualitative interview study was part of the project "Patient Experiences with Multiple Sclerosis" (PExMS) aiming (a) to investigate patients' experiences with the diagnosis of MS, their everyday life, and the different treatment approaches with DMTs, alternative medicine, rehabilitation, and lifestyle adjustments and (b) to create a website based on these patients' experiences (for more information on the PExMS project see [37]). The focus of the first group of patients which we recruited for the project based on the idea of providing especially experiences with different immunotherapies were patients with RRMS. The current substudy of this group investigates patients' experiences with lifestyle adjustments. Patients experiences with DMTs and rehabilitation are part of different analyses. For this study, a maximum variation sampling strategy [38] was applied to gather heterogeneous experiences with different MS therapeutic approaches. Participants were recruited from MS support groups, clinics, and rehabilitation centers. The inclusion criteria were (a) age $\geq$ 18 years and (b) a diagnosis of RRMS. PwMS with PPMS, poor knowledge of the German language, and severe cognitive impairments based on clinical impression were excluded from the study. Notably, the transition of RRMS to SPMS often covers a period of uncertainty, which can lead to difficulties in reliable distinction of RRMS and SPMS [39]. Hence, patients who were initially considered

RRMS were identified as rather SPMS during the interview process. However, they were kept and questions referred to their experiences when having RRMS.

## Data collection

Demographic data including MS-related information and data on self-reported severity of disability were collected. The latter was assessed using the "Patient Determined Disease Steps" (PDDS) scale, which contains ordinal levels from 0 (normal) to 8 (bedridden) [40]. Qualitative data were collected by means of problem-centered, audio and video recorded interviews according to Witzel et al. [41] including mainly open questions on patients' experiences of coping with the diagnosis of MS, of how to live with MS in everyday life, and of different treatment approaches (S1 Appendix). The interview guide was created in collaboration with an advisory board consisting of neurologists, representatives of pwMS and researchers (neurology, psychology, medical sociology, health sciences), and an expert panel on qualitative methods at the University Medical Center Hamburg-Eppendorf. A pilot run with five interviewees was performed to check for the comprehensibility of the questions and an adequate duration of the interview guide. The five interviews were fully included in the study sample without further adjustment to the interview schedule. All interviews were conducted by a single interviewer (A.S.)–a female health scientist with expertise in qualitative research methods, who had no relationship to the patients in our study. To include pwMS from all over Germany, and those with impaired mobility, the interviews were carried out in different locations on interviewees' choice like participants' homes and workplaces, clinics, rehabilitation centers, or hotels. Interviews ranged from 20 to 97 minutes (mean 45.6 minutes). Audio- and videotapes were converted into verbatim transcripts. Participants received an incentive of 20 €.

## Data analysis

Data were analyzed inductively and deductively according to the six-step (reflexive) thematic analysis of Braun and Clarke [42–45] by two researchers (S.EW., A.S.) using the software program MAXQDA Analytics Pro 2018. After familiarizing with the whole dataset (step 1), initial codes were generated (step 2). These codes were further examined and grouped into potential major themes and sub-themes according to their thematic consistency (step 3). In the next step, a thematic map of the analysis was created to ensure coherence within and distinctness between the themes (step 4). Furthermore, this step included reflection and refinement by the coders and the previously mentioned research team. Step five involved the clear definition and naming of the themes. The sixth step included the reporting of analysis results. An English/ German bilingual translated the quotations used in this article. To ensure comprehensive reporting of the study, we followed the consolidated criteria for reporting qualitative research (COREQ) [46] (S2 Appendix).

## Results

### Sample

There were 50 pwMS who participated (Table 1). Five interviews were performed in Eastern Germany, 23 in the South and 22 in the North of Germany.

### Major themes

We identified three major themes: 1) nutrition and supplements, 2) exercise and physical activity, and 3) stress management. For each theme, the sub-themes "starting new habits", "maintaining previous habits" and "practiced lifestyle habits" were identified. Starting new

**Table 1. Demographic and MS-related characteristics of participants.**

| Characteristic | N | Percent |
|---|---|---|
| Females | 35 | 70% |
| Age (mean, range) | 44.4 | 21–61% |
| Highest professional qualification | | |
| Still in vocational training | 1 | 2% |
| No professional qualification | 2 | 4% |
| Vocational education | 27 | 54% |
| Bachelor's degree | 4 | 8% |
| Master's degree, Diploma, state examination | 14 | 28% |
| Doctorate | 2 | 4% |
| MS type | | |
| RRMS | 44 | 88% |
| SPMS | 6 | 12% |
| Current DMT use | 39 | 78% |
| | M | Range |
| Years with MS since diagnosis | 13.4 | 2–33 |
| Patient determined disease steps (PDDS) | 2.7 | 0–7 |

habits was defined as lifestyle adjustment from the time of diagnosis, whereas maintaining previous habits meant the preservation of pre-diagnosis behavior. Various practiced lifestyle habits were reported by pwMS (S3 Appendix) and are shown in Figs 1–3, respectively.

**1) Nutrition and supplements.** Fig 1 gives an overview of the sub-themes of nutrition and supplements.

*Starting new dietary habits.* Almost all patients were concerned with their diet and reported various reasons to start new forms of diet. Some decided on new dietary habits after research for **information** on social media platforms or **advice** from family and friends. Other pwMS opted for nutritional counselling by professional nutritionists or searched for information on current research. Some pwMS reported, that neurologists recommended

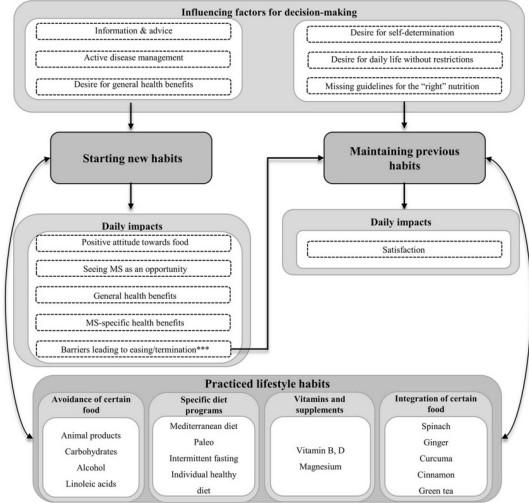

**Fig 1. Sub-themes of nutrition and supplements.** *** Barriers arising while starting new habits resulted in the resumption of previous habits prior to the diagnosis of MS.

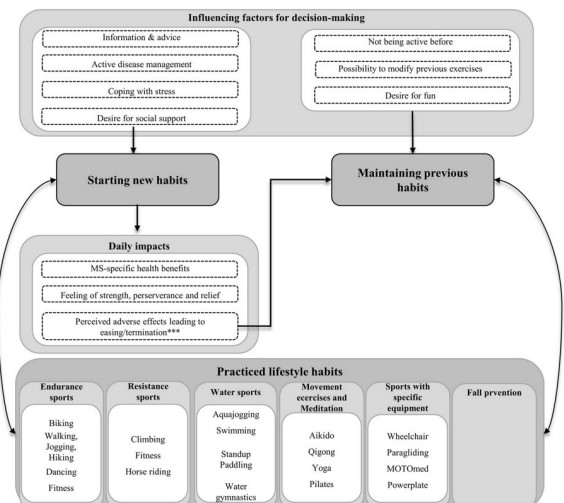

**Fig 2. Sub-themes of exercise and physical activity.** *** Perceived deterioration while starting new habits resulted in the resumption of previous habits before the diagnosis of MS.

supplementation with vitamins: *"What I additionally use in terms of medication, also from a medical point of view, [are] vitamins, vitamin B and vitamin D supplements [. . .] which are supposed to somewhat counteract the fatigue."* (pwMS 25). Furthermore, pwMS felt that they could **actively manage their disease by selecting certain types of food**, because they had observed a deterioration of pre-existing MS symptoms in association with specific food: *"[I] ate something [. . .] in cream sauce [. . .], afterwards I had numb hands, completely and utterly. [. . .] I've always had the feeling [. . .] that symptoms are worse or less bad depending on what I eat."* (pwMS 36). Additionally, pwMS felt a general need to take a more active part in the management of their disease and therefore started a new diet: *"I thought I have to do something to give myself the feeling that I was doing something. [. . .] just this feeling of sitting there and doing nothing, and to wait for everything to get worse, that was something I [. . .]*

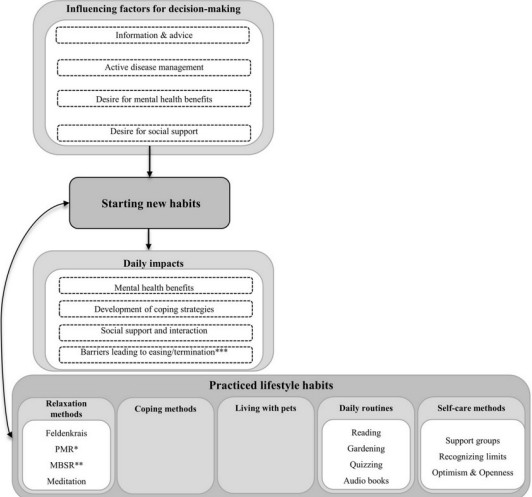

**Fig 3. Sub-themes of stress management.** * PMR = Progressive Muscle Relaxation according to Jacobson; ** MBSR = Mindfulness-Based Stress Reduction.

*couldn't stand very well."* (pwMS 36). Some pwMS integrated particular more or less scientific findings regarding diet and MS into their disease and treatment concepts: *"My diet consists of low linoleic acid food, because these very linoleic acids can be blamed for destroying the myelin layer"* (pwMS 50). Most pwMS who opted for a diet low in animal fat or sugar, aimed to achieve anti-inflammatory effects: *"Since February, I've been trying to avoid sugar so that the [. . .] sugar [. . .] doesn't feed inflammation."* (pwMS 40). Health improvement was not only sought with regard to MS-specific effects, but also in the context of **general health benefits,** such as a desired weight loss.

The implementation of new dietary habits had different **impact** on the participants' lives. Having started on a new diet, patients developed a **positive, more conscious, and appreciative attitude towards food**. Other pwMS saw **MS as an opportunity** for positively changing their eating habits: *"Well, the diagnosis helped me to start a sensible diet."* (pwMS 20). Furthermore, **general health benefits**, for example weight loss, were noted: *"I also lost thirty kilos now, because of the diet and because of the sports that I then began as well."*(pwMS 06). Similarly, general well-being was reported to improve with a certain diet: *"Now I feel a lot better than before."* (pwMS 45). PwMS described **MS-specific health benefits**, for example a reduction of relapses after the avoidance of nicotine, animal fats and alcohol. One patient speculated if the onset and course of MS might be related to dietary habits: *"I can't judge whether this has a positive or negative influence on MS, because I don't know what it would look like if I didn't do it. The only thing I know is that when it was left the old way, MS developed [. . .] and evolved."* (pwMS 09).

Other pwMS reported **barriers** when trying to integrate completely new dietary habits into their daily routine. Sometimes these barriers led to a discontinuation of the new dietary behavior, especially if the diet required strict adherence: *"In the beginning [. . .] I tried to change my diet completely. But I think that was too radical and that's why it didn't work. And then I just went back to old habits a bit."* (pwMS 45). Furthermore, one pwMS felt uncertain as to what she might truly achieve by the chosen diet due to the lack of generally accepted guidelines: *"I never had anyone or anything to support my theories, which were only experiences. There was nothing that would have told me: Yes, nutrition and MS, there is a connection."* (pwMS 36). For other pwMS practicing a specific diet created negative emotions such as unhappiness, guilt or the fear of worsening when making exceptions: *"You're not really allowed to eat that now. And when I ate it, it was immediately like, what does this do to me now, does this trigger something now?"* (pwMS 06). In some pwMS, the expected MS-specific health benefits did not occur, and therefore, they decided to ease their specific diet: *"At some point I had my control MRI appointment and had a new inflammation. [. . .] So somehow it didn't really help. And that was the trigger for me to go about it in a more relaxed manner."*(pwMS 06).

*Maintaining previous dietary habits.* Some interviewees already followed a balanced diet, others did not feel the necessity to change their current–unhealthy–dietary habits. As an explanation, pwMS claimed the desire for **self-determination** regarding specific dietary habits: *"I think the illness already determines so much of your life. I'm not going to be told what I should eat or what I shouldn't eat, or when I should do what kind of sports. I still decide that myself."* (pwMS 10). Others claimed the **desire to live a daily life without further restrictions**: *"It limits you. Whenever there are parties or something like that, you are always the one who has to say no and who is always left out somehow. You somehow exclude yourself a little bit, too."* (pwMS 44). In one pwMS **missing guidelines** for the "right MS diet" were the reason to decide against dietary changes: *"There's just a little lack of a good approach that you could use as a person with MS, and to say, here's your guideline. [. . .] I also think it's difficult to somehow make a guideline for everyone. Because there is not going to be a one-size-fits-all guideline."* (pwMS 45). For some, maintaining previous dietary habits created relief and **satisfaction**. For example, one pwMS

felt satisfied when treating himself to fast food: *"It may not be something that all people understand, why my visit to McDonald's is associated with quality of life, but for me that's the case."* (pwMS 13).

**2) Exercise and physical activity.** Fig 2 summarizes the sub-themes of exercise and physical activity.

*Starting new exercise and physical activity habits.* Some pwMS started with physical activity due to **information and advice** derived from social and print media. One patient decided to integrate physical activity into his daily routine after having participated in a research project: *"[I] also took part in a sports study a few years ago. And since then, [. . .] I try to exercise as regularly as possible."* (pwMS 19).

The most frequent motivation for starting physical exercise was the wish for **active disease management**: Some patients started exercising to improve their general fitness to better cope with MS symptoms and prevent further disease activity *"But then came the day when I thought to myself, that makes sense somehow. If you once more have to go to the hospital [. . .] while being weak, it would indeed be nice if you somehow had a fitter basis, so that you don't completely collapse and have to start again from scratch. Well, then the jogging started."* (pwMS 06). Some patients hoped to prevent further immobility by using specific exercising devices or by practicing Yoga. Furthermore, pwMS aimed to actively decrease symptoms such as spasticity and imbalance by integrating physical activity into daily life. Another pwMS participated in a teaching course enabling participants to avoid and handle falls: *"You should practice falling so that in the situation when it happens, you can roll off a bit better. And it is very important to be able to get up again."* (pwMS 03). Another motivation for physical exercise was the **desire to better cope with distress**: *"So, effectively for stress management, well, that's where I actually have sports."* (pwMS 50). Furthermore, the desire for **social support** by establishing and maintaining social contacts was specifically mentioned as an important reason to do sports with other pwMS, such as wheelchair sports or Aikido: *"You are more or less amongst like-minded people and can exchange ideas."* (pwMS 01).

One frequently mentioned **daily impact** of physical exercise was a **feeling of strength, perseverance, or relief** from stressful life-events and distraction from MS: *"Then you have also developed a certain discipline and a certain strength, when you want to achieve this and that. And that helps with such an illness."* (pwMS 37). Many pwMS reported **an impact on MS-specific health** upon integrating physical exercise into their activities of daily living: As an **MS-specific health benefit,** sports seemed to improve body sensation: *„Well, I always had the impression that [climbing] is really good for me, because it simply has a lot to do with coordination and with concentration, especially. [. . .] It's actually more about movement intelligence and yes, that's actually ideal for MS."* (pwMS 36). Furthermore, physical exercise may improve the stability of gait. One pwMS having participated in a teaching course enabling participants to avoid and handle falls reported a substantial increase of self-efficacy by having regained to get up again by herself: *"But I can now manage to get up again on my own, that is to say, to lift myself up, and then to stand on my legs, then straighten my upper body."* (pwMS 03). Another pwMS gave thought on the advantages but also on some disadvantages of physical exercise on MS symptoms: *"Sport helps me a lot with MS. I have good musculature, so I can still move relatively well for my circumstances. But of course, I also have a lot of mass for cramps because of all the muscles. Spasticity. [. . .] You can't get one without the other."* (pwMS 37). Other pwMS experienced **adverse effects** and therefore discontinued physical exercise: *"But when I noticed that the symptoms of recent relapses became stronger under strain, that really distressed me, that scared me and I then felt less and less well. And then, when you start crying on the treadmill because your legs feel wrong, because your arm feels funny, then that's not it."* (pwMS 49).

*Maintaining previous exercise and physical activity habits*. One important **motivation** to maintain previous exercise and physical activity habits was due to the fact that these activities had already been part of the daily routine prior to the MS diagnosis. However, several pwMS had to **modify** their physical exercise routines because of specific neurological impairments. For example, riding a regular bike was changed to using an e-bike or exercising on a bike ergometer. Similarly, hiking routes were adjusted to the individual capabilities: *"You don't do a ten-kilometer tour a day, but only three kilometers, with breaks accordingly. And you take the cable car up instead of walking up or down."* (pwMS 30). For some pwMS, having **fun** played a decisive role in continuing sports to the same extent prior to the diagnosis. One patient maintained his habit of **not being active**, because he assessed the threshold for starting physical activity as too high: *"I was the little fat one at school and sports was no fun all of my life. Taking up sports only after the diagnosis is a double challenge, simply because exercise [. . .] has never been much fun for me and now [exercise] is even limited. That means it's a rough ride. And I haven't made the journey—yet."* (pwMS 13).

**3) Stress management.** All pwMS reported new stress management habits when diagnosed with MS (Fig 3). For the sake of completeness: pre-diagnosis stress management habits were not mentioned in the interviews.

*Starting new stress management habits*. One **influencing factor** affecting the decision to change a specific habit was scientific **information** derived from current research. Another significant factor was the **advice** from relatives. **Active disease management** was an important reason to start new habits: Patients perceived an interdependency between stress levels and intensity of MS symptoms. This encourages them to change their stress management habits: „*High stress level [. . .] immediately has a physical effect on me. [. . .] I then stumble again, my dizziness immediately appears, my hands are even more numb. [. . .] Everything that has been there before [. . .] flares up. [. . .] For me, that is always immediately the sign [. . .] to somehow put boundaries around me, hand things over and try to get periods of rest for myself."* (pwMS 46). Other pwMS hoped that their relatives might facilitate coping with the diagnosis of MS: *"When we realized that my wife [. . .] was also confronted with fears due to the diagnosis, we again turned to the German Multiple Sclerosis Society and were given the opportunity to undergo couples therapy."* (pwMS 13). In addition, the **desire for mental health benefits** was often mentioned as the main motive for taking specific measures such as psychotherapy to reduce depressive symptoms and anxiety that had emerged since or with the diagnosis of MS. The **desire for social support** emerged as another important motive to deal more openly with the diagnosis of MS. Social interaction is seen as active protection against negative role assignment (victims) or even stigmatization: *"If you don't say that right out and also give the reason behind it, then you put yourself in a very strange role as a victim, because then the pity comes."* (pwMS 26), *"It has to be said that the situation is such that I have multiple sclerosis, only then my counterpart can respond to that without thinking: 'He must be lazy.'"* (pwMS 13).

From the interviews, different **impacts** of stress management can be reduced. For example, **mental health benefits** were a commonly reported effect. Relaxation was achieved by daily routines, such as listening to audio books regularly. Furthermore, setting boundaries resulted in emotional relief: *"When I notice that I'm running short of time for something, I call and say, "I don't know if I'll make it on time". And then there's usually enough time, but there's no pressure on me anymore."* (pwMS 42). Psychotherapy helped pwMS to **cope better** with the disease: *"And that's the way to deal with it, to arrive within this illness, to say: Okay, I'm not looking for the possibility of not having the illness, but I have it now and it won't go away, it will be there all my life, it's part of me. And I think psychotherapy is very important for dealing with that. Well, it sure helped me."* (pwMS 22). As another example, the Mindfulness-Based Stress Reduction technique was reported to facilitate coping with MS: *"In just dealing with it when symptoms*

appear [. . .], well, meditation has helped me a lot [. . .] not to panic directly, to perceive it first, to feel one's body somehow or to try to feel it.* (pwMS 36). Some pwMS reported that meditation helped them to accept the current situation and to be grateful for the things that work out. Others reported better coping by exchanging information in MS support groups: *"You exchange ideas about what is good, because the others have other deficits or you know more, because you've been around longer."* (pwMS 28). Several pwMS changed their perception of MS by adopting a more open and optimistic attitude thereby increasing mindfulness and health behavior: *"And I can also see something positive in MS. I think it was also a cry for help from my body, to take care of myself. And yes, to also work on myself."* (pwMS 34). Other pwMS reported changes in **social support and interaction** by making new friends and improving social behavior: *"Other people have also said to me: You have actually become a better person. I also became more sensitive myself. Today, I can also listen better."* (pwMS 22).

Some pwMS uttered **barriers** such as unrealistic expectations that led them to stop going to psychotherapy: *"Yes, I just went there: 'Well, if you can't make my illness go away, then you can't help me either'."* (pwMS 02). Other pwMS stopped attending meetings of support groups having gained unfortunate experience at some of the meeting**s**: *"The people were much older than me and, unfortunately, some of them were already very limited. That actually scared me at that moment rather than helped me."* (pwMS 43). In one pwMS, Progressive Muscle Relaxation according to Jacobson resulted in increased spasticity: *"You're supposed to strain the muscle and then let it go again, that doesn't work at all, then for me it just closes up. So spasticity is a symptom that is very sensitive to stress. So the more stress you have, the more you exert yourself, the stronger the spasticity becomes, and that's when you actually need relaxation, and for me, this was just tension."* (pwMS 28).

## Discussion

This study provides insight into experiences of pwMS with lifestyle adjustments regarding nutrition and supplements, exercise and physical activity, and stress management habits. We focused on individual factors affecting the decision of pwMS to start new lifestyle habits after having been diagnosed with MS or to maintain previous behavior. Furthermore, we aimed to understand the effects of lifestyle adjustments on daily life as experienced by pwMS.

One theory analyzing behavior change for people with chronic diseases is the Health Action Process Approach (HAPA). In this concept, two different phases are postulated: First, the motivation phase, in which people develop their intentions to change the behavior, and second, the volititonal phase, in which the intentions must be translated into action. Different factors contribute to intention formation in the motivation phase: Perceived task self-efficacy, risk perception and outcome expectations all of which can be influenced by acquiring new information. In the volitional phase, action planning and control, coping with barriers, social support and maintenance/recovery self-efficacy are correlated with successful behavior change [47]. Recent research indicates that HAPA might be a useful framework for engaging pwMS to physical activity [48].

Obtaining information and advice was one major stimulating factor to start new lifestyle habits. PwMS gathered and expanded their knowledge mainly through social media and the internet as well as by their social contacts. This knowledge encouraged them to decide on lifestyle adjustments. According to HAPA model [47], knowledge expansion might help to define the outcome expectations and risk factors to start a specific lifestyle habit. Only very few reported that their motivation was mainly driven by participation in scientific projects (e.g. research studies, nutrition counseling) or based on support by physicians and health care professionals. This is in line with current research: pwMS gather medical information about MS

mainly online [49]. Another frequently mentioned motive to start new stress management, dietary, and exercise habits was the desire to be actively engaged in one's own disease management. This goal of actively improving one's health may be related to the belief in self-efficacy, defined as a perceived capacity to overcome specific concomitant circumstances [50]. Based on this concept, pwMS feel enabled to counteract symptoms or disease activity of MS by implementing new lifestyle habits into daily living [51]. Perceived self-efficacy, a predictor for motivation in health behavior change [47], is further associated with the perception of improved physical and psychological health status [52] thereby also affecting QOL [53]. Also, pwMS in this study opted for changes in diet and stress management after having noticed an interrelation between certain types of food or stressful life events and MS symptoms or pathogenesis. To our knowledge, these relations have not yet been fully elucidated by current research. However, the unpredictability and heterogeneity of MS symptoms and the risk of disease progression is a constant challenge for pwMS when trying to create coherence and gain control over MS. Therefore, pwMS may tend to causally attribute relapses or worsening of the disease to their behavior to restore sense-making [54] and to search for a theoretical basis which may justify specific forms of diet, stress control and overall lifestyle choices. Moreover, active disease management includes a mental process of consciously dealing with MS and its consequences. Therefore, pwMS in this study started new stress management habits to develop coping strategies for both themselves and their family members. On the one hand, social support from family members is highly important for coping with MS [55]. On the other hand, however, the MS related challenges for family members are affecting their emotional well-being, their individual needs and their coping strategies [56, 57]. These should be addressed in psychoeducational counseling in a timely manner. Social support is correlated with successful behavior change according to the HAPA concept [47] and was another concise motivator for lifestyle adjustments in our study. PwMS joined sports or support groups to obtain peer group support. Social support, especially from peers, family and friends, is associated with adequate coping with MS [55].

PwMS in this study reported different effects of new lifestyle habits on daily living. After having implemented specific exercise and dietary adjustments into daily life, most pwMS in this study mainly noticed improved symptom control. While the effects of dietary changes on MS symptoms and disease progression are discussed controversially due to the low-quality evidence of trials [9], the positive effects of physical activity on MS symptoms are well established. Hence, the reported effects of physical activity in this study such as improved balance, mobility and stress reduction are well in line with scientific evidence [15, 16]. In general, most pwMS in this study improved coping with MS through stress management techniques and, interestingly, dietary changes: Some interviewees in this study changed their attitude towards MS by cognitive restructuring, psychological interventions, meditation, and healthy dietary changes. Thereby, they recognized MS as a source for increased mindfulness and improved self-care. These mental adjustments towards a perception of benefits from a given disease and towards a more optimistic view on MS are part of the concept of positive psychology and associated with increased mental health and well-being [58], QOL [59] and perceived control over MS [60]. These findings indicate that psychological and cognitive approaches are important for the coping with MS and QOL—regardless of the physical status. Although social support is a critical factor in positive coping with MS [55], in this study the perception of social support is highly individual: For some pwMS, participation in support groups was experienced as supportive, whereas for others, negative emotions prevailed. The latter finding is in accordance with an Italian survey by Uccelli et al. in which no beneficial outcome of a support group program was observed. Even more, the authors stated that mentally healthy pwMS may be at risk for psychological deterioration when joining support groups [61]. On the other hand, it was

demonstrated in a study from the UK including 152 pwMS, that mental well-being was significantly associated with support group identification [62], and in a German study with 1220 pwMS, that support groups were not associated with QOL, but several other aspects of well-being such as health literacy and self-management [63, 64].

Some pwMS discontinued physical activity due to aggravation of pre-existing symptoms and attributed physical impairment and pain to exercise adjustments. However, these causal relations are not supported by current literature [65, 66]. On the contrary, there is no evidence that exacerbations are caused by physical exercise [65]. As for nutritional adjustments, negative emotions and strict adherence to the diet made it difficult for some pwMS in this study to stick to their new dietary concepts. This is in line with a qualitative study addressing self-management procedures in people with chronic disease (e.g. MS, diabetes, rheumatism). The authors concluded that lifestyle habits can only be successfully integrated into daily living in the long run as long as they do not interfere with other important activities in everyday life [67]. These findings indicate the need for a balanced lifestyle concept, so that pwMS feel self-efficient, but not restricted or overburdened. In this study, some pwMS discontinued their newly integrated lifestyle habits because their expectations of remaining relapse-free or free of progression were not met. According to the social-cognitive theory, outcome expectations have a direct effect on health-promoting behavior [68]. Therefore, realistic information on how lifestyle factors may affect MS-specific health is urgently needed to ensure compliance with lifestyle adjustments.

Accordingly, the lack of evidence-based data and information was reported frequently in our interviews: Missing guidelines prevented some pwMS from starting on new dietary habits and the uncertainty about the clinical effects was the main reason to prematurely stop them. Current literature supports the unmet need for information on effective lifestyle habits for pwMS [36, 69]. This lack of information is all the more unfortunate since pwMS consider physicians' professional recommendations as most trustworthy [49] and pwMS have the capacity to fully understand scientifically based treatment effects and to incorporate them into their decision process [70, 71]. These results strengthen the need for evidence-based research on the effectiveness of lifestyle adjustment and further underline health professionals' responsibilities to communicate the evidence to their patients and promote shared decisions on lifestyle adjustments.

Other pwMS in our interviews decided to maintain the pre-diagnosis lifestyle, because they did not want MS to take control over their lives. This wish for self-determination is of critical importance, since willful ignorance may result in decreased motivation [72]. For lifestyle maintenance, enjoyment was another motive to keep the same level of physical activity and dietary habits that were practiced prior to the diagnosis of MS, which is in line with the results of another qualitative study investigating psychosocial adjustment to MS [73]. PwMS have to deal with the uncertainties associated with the course of MS and are continually forced to adapt to physical, psychological and relationship changes [74]. Focusing on one's current needs may therefore be one way of dealing with the sometimes unpredictable clinical course of MS. To get involved in physical activity, the pre-diagnosis lifestyle played an important role for pwMS in this study: For patients who had never followed a healthy lifestyle, the psychological obstacle to change their attitude was hard to overcome. A qualitative study investigating self-management strategies in people with chronic diseases (e.g. MS, diabetes, rheumatism) demonstrated that growing up on a healthy lifestyle, facilitates the permanent integration of health-promoting habits [67].

In summary, this study demonstrates high relevance of dietary habits, physical activity, and stress management habits among pwMS. Many of the influencing factors and outcomes reported in this study are mirrored in Bravo's patient empowerment model, which describes various indicators, ethos, moderators and outcomes of patient empowerment in people with chronic diseases. For example, self-determination, sense of self-efficacy, experiencing

control and coherence in dealing with the disease, active disease management, information-seeking, and shared decision-making affect patient empowerment, which in turn leads to improved QOL and well-being and potentially better health outcomes [75]. In general, motives for lifestyle habits and their mental and physical MS-specific effects varied widely between individual pwMS in this study. During the decision-making process and the integration of new lifestyle habits, each pwMS faces various problems along with heterogenous symptoms and disease progression [76]. To deal with these unique circumstances, the aforementioned adjustments inevitably evokes the evolvement of individual preferences, needs, and coping strategies. Unlike the evidence-based guideline for pwMS concerning physical activity [77], there are no evidence-based recommendations for stress management and nutrition. This lack of scientific evidence facilitates the adherence to lifestyle adjustments that are not based on scientific evidence [30].

## Limitations

This study is part of a larger qualitative study examining, among other things, pwMS' experiences with DMTs [78]. Data collection focused on DMTs rather than lifestyle adjustments. In particular, it would not have been impossible to define a saturation endpoint because of the many individual lifestyle choices. RRMS is the most common course of MS with the largest diversity of treatment and management options. For better comparability, RRMS was set at inclusion criterium and patients with PPMS were excluded from the actual study. Besides, the recruitment of pwMS who consented to video and audio recording was difficult. For this reason, the current sample may differ from the "average" MS population in favor of pwMS who are more open and motivated and therefore more willing to engage in new lifestyle adjustments. Furthermore, mainly open questions about lifestyle experiences in general were asked, rather than inquiring specifically about particular lifestyle habits. For example, smoking was rarely mentioned by our interviewees and therefore not included in this study. Nevertheless, smoking should be considered an important lifestyle habit for pwMS because smoking affects both disease susceptibility and disease progression [5, 8, 79] and quitting smoking may reduce disease progression [5, 80].

## Conclusion

This study provides rich and nuanced information on patients' experiences with different lifestyle habits. These findings demonstrate the complexity and heterogeneity of influencing factors in the decision-making process. Positive and negative effects of lifestyle adjustments on health are reported. The heterogeneous symptoms and sometimes unpredictable course of MS, the pre-diagnosis lifestyle habits, and the social surroundings of pwMS need to be taken into account when offering adequate, sufficient and personalized counseling.

 Three main conclusions may be drawn from the data presented here: The results presented here lead to three important conclusions: 1) Further research is warranted to better define the effects of lifestyle habits on MS symptoms and progression, in particular with regard to diet and stress reduction 2) Patient education by health professionals in MS should include the available evidence (e.g. physical activity). 3) Patients who wish to undergo lifestyle adjustments need to be actively supported in changing their behavior (preferably through e-health interventions).

## Supporting information

**S1 Appendix. Interview guide.**
(DOCX)

**S2 Appendix. COREQ (COnsolidated criteria for REporting Qualitative research) check-list.**
(DOCX)

**S3 Appendix. Themes, sub-themes and the corresponding exemplary quotes from pwMS.**
(DOCX)

## Acknowledgments

First, we thank all pwMS who have made this research possible. Furthermore, the support of the members of our working group for patient autonomy in this qualitative interview study is greatly appreciated. We are grateful for the assistance of Desiree Eklund as part of the advisory panel.

## Author Contributions

**Conceptualization:** Saskia Elkhalii-Wilhelm, Anna Sippel, Karin Riemann-Lorenz, Christopher Kofahl, Jutta Scheiderbauer, Sigrid Arnade, Ingo Kleiter, Stephan Schmidt, Christoph Heesen.

**Data curation:** Saskia Elkhalii-Wilhelm, Anna Sippel.

**Formal analysis:** Saskia Elkhalii-Wilhelm, Anna Sippel.

**Funding acquisition:** Christoph Heesen.

**Investigation:** Saskia Elkhalii-Wilhelm, Anna Sippel, Christoph Heesen.

**Methodology:** Saskia Elkhalii-Wilhelm, Anna Sippel, Karin Riemann-Lorenz, Christopher Kofahl, Jutta Scheiderbauer, Sigrid Arnade, Ingo Kleiter, Stephan Schmidt, Christoph Heesen.

**Project administration:** Anna Sippel, Christoph Heesen.

**Supervision:** Anna Sippel, Karin Riemann-Lorenz, Jutta Scheiderbauer, Sigrid Arnade, Ingo Kleiter, Stephan Schmidt, Christoph Heesen.

**Validation:** Saskia Elkhalii-Wilhelm, Anna Sippel, Christoph Heesen.

**Visualization:** Saskia Elkhalii-Wilhelm.

**Writing – original draft:** Saskia Elkhalii-Wilhelm, Anna Sippel.

**Writing – review & editing:** Karin Riemann-Lorenz, Christopher Kofahl, Jutta Scheiderbauer, Sigrid Arnade, Ingo Kleiter, Stephan Schmidt, Christoph Heesen.

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
