## [Decision Letter · Decision Letter 0]

1 Dec 2021

PONE-D-21-10862Experiences of Persons with Multiple Sclerosis with Lifestyle Adjustment - A Qualitative Interview StudyPLOS ONE

Dear Dr. Elkhalii-Wilhelm,

Thank you for submitting your manuscript to PLOS ONE. After careful consideration, we feel that it has merit but does not fully meet PLOS ONE’s publication criteria as it currently stands. Therefore, we invite you to submit a revised version of the manuscript that addresses the points raised during the review process.

I have added some comments below. Most notably the methods section needs development. 

We look forward to receiving your revised manuscript.

Kind regards,

Andrew Soundy

Academic Editor

PLOS ONE

Additional Editor Comments (if provided):

Thank you for your submission. Please develop your methods section to identify all elements found in a qualitative framework article e.g., https://pubmed.ncbi.nlm.nih.gov/24979285/

Journal Requirements:

2. Please describe in your methods section how capacity to consent was determined for the participants in this study.

"AB has received funding from Roche. CH has received research grants, congress travel compensations and salaries for talks from Biogen, Genzyme, Sanofi-Aventis, Bayer Healthcare, Merck, Teva, Roche, and Novartis. IK has received speaker honoraria and travel funding from Bayer, Biogen, Novartis, Merck, Sanofi Genzyme, Roche; speaker honoraria from Mylan; travel funding from the Guthy-Jackson Charitable Foundation; consulted for Alexion, Bayer, Biogen, Celgene, Chugai, IQVIA, Novartis, Merck, Roche; and received research support from Chugai, Diamed. SEW, KRL, CK, JS, SA and SS declare having no competing interests. "

We note that you received funding from a commercial sources: Biogen, Genzyme, Sanofi-Aventis, Bayer Healthcare, Merck, Teva, Roche, Novartis, Celgene, Chugai and IQVIA.

5. Please amend your list of authors on the manuscript to ensure that each author is linked to an affiliation. Authors’ affiliations should reflect the institution where the work was done (if authors moved subsequently, you can also list the new affiliation stating “current affiliation:….” as necessary).

6. Your abstract cannot contain citations. Please only include citations in the body text of the manuscript, and ensure that they remain in ascending numerical order on first mention.

Reviewers' comments:

Reviewer's Responses to Questions

**Comments to the Author**

1. Is the manuscript technically sound, and do the data support the conclusions?

Reviewer #1: Yes

Reviewer #2: Yes

2. Has the statistical analysis been performed appropriately and rigorously? 

Reviewer #1: N/A

Reviewer #2: N/A

3. Have the authors made all data underlying the findings in their manuscript fully available?

Reviewer #1: Yes

Reviewer #2: No

4. Is the manuscript presented in an intelligible fashion and written in standard English?

Reviewer #1: Yes

Reviewer #2: Yes

5. Review Comments to the Author

Reviewer #1: Thank you for giving me the opportunity to read and comment the article “Experiences of Persons with Multiple Sclerosis with Lifestyle Adjustment - A Qualitative Interview Study”.

The objective of the article is to describe the experiences of people with Multiple Sclerosis with lifestyle adjustments. Patients’ management of the disease and adjustment to the new life after the diagnosis of MS is an important topic and the current article gives interesting insights.

The introduction gives a clear overview of the topic. It should be noted that some studies did focus their attention on the experience of people with MS with lifestyle adjustment; for example Neate et al. (2021). (DOI:10.1111/hex.13364) explored perspectives of pwMS regarding the modification of lifestyle-related risk factors in multiple sclerosis and Smith et al. (2019) (DOI: 10.1002/pri.1776) investigated key factors that influence participation in sport and exercise. The authors might verify if the statement in lines 98-99 should be modified according to the existing literature.

In the methods section (lines 110-113), the current study is presented as a part of a larger study (the PExMS project). However, it is not entirely clear how the study is connected to the PExMS project. I think the reader would benefit of a more detailed explanation of the purpose of the current study and its rationale in the broader project.

In the method section it is stated one inclusion criteria is a diagnosis of RRMS and that PwMS with a progressive course of MS were be excluded from the study (lines 116-117). It would be useful for the reader to have an explanation of the reason for this choice. Moreover, in the results (table 1) it appears that 6 participants have a diagnosis of SPMS. Could the authors clarify this matter?

Quotations provided in the results are very clear and help explain the themes.

I think that some discrepancies between the designated objectives and the reported results might be present. E.g., at line 100 decision-making process concerning different lifestyle adjustments is introduced as one of the main fields that the study aims to explore, but it is not properly covered within the following discussion. The focus seems to be more on the influencing factors, which of course are a fundamental element of the decision-making process but does not fully cover all the steps and psychological aspects necessary to make a change. Moreover, the third sub-theme mentioned at line 159 (“practiced lifestyle habits”) does not seem to have a correspondence in the results: is that and the voice “maintaining previous habits” the same thing? I suggest maintaining linearity between objectives and results so that the reader can better understand the underlying thought process of this study.

As reported in the limit section, some data were excluded from analysis (lines 492-493) but this does not seem to be in line with the objective of the research which was to “explore the experiences of pwMS with different lifestyle adjustments”. Reading the results I had the impression that patients mentioned only the three themes presented (nutrition and supplements, exercise and physical activity, and stress management) and, therefore, that patients included in the study did not consider other lifestyle adjustments important. The exclusion of some data based on the times they were mentioned might mislead the reader, as it was a choice of the authors that is not declared until the limit section. I would suggest to better explain this choice in the method section or to consider adding another theme to reflect the complexity of the data (for example “Other lifestyles”). This might contribute to have a higher consistency between objective and results.

Reviewer #2: This is a qualitative study where the researcher interviewed a large number of people with MS (n=50) around their lifestyle choices and adjustments they made. It is an important area of research that we don't know much. However, I have several reservations that I will try to explain below:

Abstract

1. The conclusions need to be more specific, with specific take-away messages for the reader.

2. In the abstract, it is reported that only people with relapsing-remitting MS were interviewed, but on the methods, it says that people with secondary progressive MS were also included.

Introduction

3. The rationale for excluding people with primary progressive MS is not clear.

4. From the introduction, I understand that the evidence linking lifestyle adjustment and impact on MS relapses or progression is not strong, with only preliminary data and pilot studies reported in most cases. It would have been helpful in the introduction when discussing lifestyle adjustments to underline whether we have strong or weak evidence about their effectiveness.

Methods

5. Were there any changes in the interview schedule after the first five pilot interviews?

6. The topic guide does not align with the aims of the study. Only one question (4. Apart from DMTs, there are other therapies that can be used. These include alternative therapies or measures that change lifestyle habits.) discusses the lifestyle adjustments partly.

7. In the limitations, it is mentioned that this research project is part of a larger research project. Most transparency is needed in the methods section about the aims of the bigger project, how this study fitted within the larger study and how data were extracted.

8. Again in the limitations, it is mentioned that the interviews were audio and video recorded, but this is not reported in the methods section.

9. The reporting of the study did not follow COREQ guidelines. Mainly there are missing criteria from Domain 1: 'Research team and Reflexivity'.

Results

10. It is mentioned that each theme has 3 sub-themes, but I can't see the third subtheme, 'practiced lifestyle habits', in any of the themes.

11. The results section is very interesting and informative. However, I wonder whether re-arranging the theme will enable more of the details to come out. Instead of looking at specific lifestyle areas (stress, nutrition, exercise), the themes could be formed around the key elements underpinning all of these lifestyle choices, e.g. control, empowerment, information from health care professionals, benefits of not changing. Focusing on these themes can bring out more useful information on lifestyle adjustment that can be taken forward in future interventions and clinical practice.

12. The title of the theme 'stress management is not accurate as the theme mainly discusses ways people use to manage their mental health more broadly and their relationships with others.

13. 'Maintaining habits' sub-theme is not discussed within the 'stress management theme.

Discussion

14. The discussion mainly repeats the findings. It would be helpful to discuss further theories and evidence around habit formation and behaviour change and how what we know about lifestyle adjustment in general fits (or not) within the MS population.

6. PLOS authors have the option to publish the peer review history of their article (what does this mean?). If published, this will include your full peer review and any attached files.

Reviewer #1: **Yes: **Silvia Poli

Reviewer #2: **Yes: **Angeliki Bogosian

---

## [Author Response · Author response to Decision Letter 0]

1 Mar 2022

Point-By-Point Reply:

Comment from Editor:

Comment 1: Please ensure that your manuscript meets PLOS ONE's style requirements, including those for file naming.

Response 1: Thank you. We carefully went through our manuscript and PLOS ONE’s style requirements and hope we have thereby met all requirements of PLOS ONE.

Comment 2: Please describe in your methods section how capacity to consent was determined for the participants in this study.

Response 2: As written in line 134 ff PwMS with PPMS, poor knowledge of the German language, and severe cognitive impairments were excluded from the study. We now added “…based on clinical impression”.

Comment 3: We note that you received funding from a commercial sources: Biogen, Genzyme, Sanofi-Aventis, Bayer Healthcare, Merck, Teva, Roche, Novartis, Celgene, Chugai and IQVIA. Please provide an amended Competing Interests Statement that explicitly states this commercial funder, along with any other relevant declarations relating to employment, consultancy, patents, products in development, marketed products, etc. Within this Competing Interests Statement, please confirm that this does not alter your adherence to all PLOS ONE policies on sharing data and materials by including the following statement: "This does not alter our adherence to PLOS ONE policies on sharing data and materials.” Please include your amended Competing Interests Statement within your cover letter. We will change the online submission form on your behalf.

Response 3: Thank you. We have included the Competing Interests Statement within our reply letter and uploaded it. As indicating by the editor all of these fundings do not alter our adherence to PLOS ONE policies on sharing data and materials.

Comment 4: We note that you have indicated that data from this study are available upon request. PLOS only allows data to be available upon request if there are legal or ethical restrictions on sharing data publicly. a) If there are ethical or legal restrictions on sharing a de-identified data set, please explain them in detail (e.g., data contain potentially sensitive information, data are owned by a third-party organization, etc.) and who has imposed them (e.g., an ethics committee). Please also provide contact information for a data access committee, ethics committee, or other institutional body to which data requests may be sent. b) If there are no restrictions, please upload the minimal anonymized data set necessary to replicate your study findings as either Supporting Information files or to a stable, public repository and provide us with the relevant URLs, DOIs, or accession numbers. 

Response 4: Thank you for this feedback. We have integrated the statement in our reply letter to explain the restrictions on sharing our data publicly.

Comment 5: Please amend your list of authors on the manuscript to ensure that each author is linked to an affiliation. Authors’ affiliations should reflect the institution where the work was done (if authors moved subsequently, you can also list the new affiliation stating “current affiliation:….” as necessary)

Response 5: Thank you for this advice. Jutta Scheiderbauer is a competent patient representative, who is not belonging to any affiliation. Thus, we have amended the list of authors as follows:

Line 13: 3 Patient representative, Trier, Germany.

Comment 6: Your abstract cannot contain citations. Please only include citations in the body text of the manuscript, and ensure that they remain in ascending numerical order on first mention.

Response 6: We are sorry for this mistake and excluded the citation.

Comment 7: Please include captions for your Supporting Information files at the end of your manuscript, and update any in-text citations to match accordingly.

Response 7: Thank you. We have included the captions at the end of our manuscript and updated the citations accordingly. 

Comments from reviewer 1:

Thank you for giving me the opportunity to read and comment the article “Experiences of Persons with Multiple Sclerosis with Lifestyle Adjustment - A Qualitative Interview Study”.

The objective of the article is to describe the experiences of people with Multiple Sclerosis with lifestyle adjustments. Patients’ management of the disease and adjustment to the new life after the diagnosis of MS is an important topic and the current article gives interesting insights.

Comment 1: The introduction gives a clear overview of the topic. It should be noted that some studies did focus their attention on the experience of people with MS with lifestyle adjustment; for example Neate et al. (2021). (DOI:10.1111/hex.13364) explored perspectives of pwMS regarding the modification of lifestyle-related risk factors in multiple sclerosis and Smith et al. (2019) (DOI: 10.1002/pri.1776) investigated key factors that influence participation in sport and exercise. The authors might verify if the statement in lines 98-99 should be modified according to the existing literature.

Response 1: We thank you for this constructive advice and have included the mentioned citations and changed as follows:

Line 108 ff: “A few studies of patients' experiences with lifestyle habits have focused perspectives about general lifestyle risk factors (34), experiences with challenges in physical activity (35) or nutrition (36) after MS diagnosis. However, deeper insight into patient experiences with regard to the variety of lifestyle behaviors, the implementation of new and the maintenance of previous specific habits, is under-researched.”

Comment 2: In the methods section (lines 110-113), the current study is presented as a part of a larger study (the PExMS project). However, it is not entirely clear how the study is connected to the PExMS project. I think the reader would benefit of a more detailed explanation of the purpose of the current study and its rationale in the broader project.

Response 2: Thank you for this helpful advice. We implemented your feedback and hope we could clarify the position of our study within the project as described in the changes below:

Lines 124 ff: “This qualitative interview study was part of the project “Patient Experiences with Multiple Sclerosis” (PExMS) aiming (a) to investigate patients’ experiences with the diagnosis of MS, their everyday life, and the treatment approaches with DMTs, alternative medicine, rehabilitation and lifestyle adjustments, and (b) to create a website based on these patients’ experiences (34). The focus of the first group of patients which we recruited for the project based on the idea of providing especially experiences with different immunotherapies were patients with RRMS. The current substudy of this group investigates patients’ experiences with lifestyle adjustments. Patients experiences with DMTs and rehabilitation are part of different analyses.”

Comment 3: In the method section it is stated one inclusion criteria is a diagnosis of RRMS and that PwMS with a progressive course of MS were be excluded from the study (lines 116-117). It would be useful for the reader to have an explanation of the reason for this choice. Moreover, in the results (table 1) it appears that 6 participants have a diagnosis of SPMS. Could the authors clarify this matter?

Response 3: The PExMS project was based on the idea to develop a resource of patient experiences for patients with MS. We chose to start with RRMS as especially drug experiences are most complex within this patient group. Lifestyle adjustments might differ according to disease courses and we hope to include also patient groups from other disease stages in the future. 

We have added the following sentences to our limitations to clarify the matter for the reader:

Lines 528 ff: “RRMS is the most common course of MS with the largest diversity of treatment and management options. For better comparability, RRMS was set as inclusion criterium and patients with progressive MS were excluded from the actual study.”

For the matter of SPMS, we have added the following sequence to our manuscript:

Lines 138 ff: “Notably, the transition of RRMS to SPMS often covers a period of uncertainty, which can lead to difficulties in reliable distinction of RRMS and SPMS (36). Hence, patients who initially were considered RRMS were identified as being rather SPMS during the interview process. However, they were kept and questions referred to their experiences when having RRMS. “

Comment 4: Quotations provided in the results are very clear and help explain the themes.

I think that some discrepancies between the designated objectives and the reported results might be present. E.g., at line 100 decision-making process concerning different lifestyle adjustments is introduced as one of the main fields that the study aims to explore, but it is not properly covered within the following discussion. The focus seems to be more on the influencing factors, which of course are a fundamental element of the decision-making process but does not fully cover all the steps and psychological aspects necessary to make a change. Moreover, the third sub-theme mentioned at line 159 (“practiced lifestyle habits”) does not seem to have a correspondence in the results: is that and the voice “maintaining previous habits” the same thing? I suggest maintaining linearity between objectives and results so that the reader can better understand the underlying thought process of this study.

Response 4: Thank you for your helpful comment on the results of our study. As cited by the reviewer we did include influential factors. And we agree that the decision-making process consists of more than influencing factors and includes especially also psychological aspects. However, psychological factors were not addressed extensively by participants. Nevertheless, we found some psychological aspects addressed by the patients that influence the decision-making process, such as the desire for self-determination (line 249), the desire to better cope with distress (line 284), the desire for social support (line 285, line 346) and the desire for mental health benefits (line 344). 

The third sub-theme “practiced lifestyle habits” is reflected in the Figures Fig 1 (referred to on page 9, line 192), Fig 2 (referred to on page 12, line 264) and Fig 3 (referred to on page 14, line 326). 

To clarify, we have added the following sentence in line 189: 

“Various practiced lifestyle habits were reported by pwMS (S2 Appendix) and are shown in Fig 1, Fig 2 and Fig 3, respectively.” 

By naming and subdividing these practiced lifestyles, we aim to illustrate the high diversity and individuality of lifestyle behaviors for our patients. As can be seen in the Figures 1-3 there is a wide variety of lifestyle habits that are covered by the three main themes. Naming and describing each of these behaviors in the result section would exceed the scope of the paper. However, we felt it was important to illustrate the various lifestyle habits in the Figures Fig 1-3 because they reflect how individually lifestyle adjustments were perceived by our patients and are therefore, in our opinion, elementary to this qualitative interview study. 

Comment 5: As reported in the limit section, some data were excluded from analysis (lines 492-493) but this does not seem to be in line with the objective of the research which was to “explore the experiences of pwMS with different lifestyle adjustments”. Reading the results I had the impression that patients mentioned only the three themes presented (nutrition and supplements, exercise and physical activity, and stress management) and, therefore, that patients included in the study did not consider other lifestyle adjustments important. The exclusion of some data based on the times they were mentioned might mislead the reader, as it was a choice of the authors that is not declared until the limit section. I would suggest to better explain this choice in the method section or to consider adding another theme to reflect the complexity of the data (for example “Other lifestyles”). This might contribute to have a higher consistency between objective and results.

Response 5: We appreciate this helpful advice. All mentioned lifestyle habits of our patients are listed in the subtheme "Practiced lifestyle habits". In our opinion, this again demonstrates the relevance of illustrating this subtheme in the Figures Fig 1-3. Only two patients mentioned having smoked in a subordinate sentence. After careful consideration, we decided not to categorize this as a "main theme" because these patients did not mention any "influencing factors" or "impacts on daily life" related to smoking. However, we agree with the reviewer’s statement that it is not consistent with the approach of qualitative research to leave a given topic (even if tackled only marginally) entirely unmentioned and therefore included the issue of smoking habits in the limitations section of the discussion (page 23, line 534). However, we did not include “Smoking” in a main theme, such as “other lifestyles”, because we did not have enough data for this category. 

Comments from reviewer 2:

This is a qualitative study where the researcher interviewed a large number of people with MS (n=50) around their lifestyle choices and adjustments they made. It is an important area of research that we don't know much. However, I have several reservations that I will try to explain below:

Comment 1: Abstract: The conclusions need to be more specific, with specific take-away messages for the reader.

Response 1: We appreciate this comment and hope we could clarify the conclusions by adding the following take-away messages to the Abstract:

Lines 49 ff: “This study provides a rich and nuanced amount of experiences of pwMS with lifestyle adjustments and leads to three important conclusions: 1) Further research is warranted to better describe the perceived effects of lifestyle habits on MS symptoms and progression, in particular with regard to nutrition and stress reduction; 2) patient education in MS should include the available evidence on lifestyle management and 3) patients need to be actively supported in changing their lifestyle behavior.”

Comment 2: Abstract: In the abstract, it is reported that only people with relapsing-remitting MS were interviewed, but on the methods, it says that people with secondary progressive MS were also included.

Response 2: See comment 3 of reviewer 1 and reply. 

Comment 3: Introduction: The rationale for excluding people with primary progressive MS is not clear

Response 3: See comment 3 of reviewer 1 and reply. 

Comment 4: Introduction: From the introduction, I understand that the evidence linking lifestyle adjustment and impact on MS relapses or progression is not strong, with only preliminary data and pilot studies reported in most cases. It would have been helpful in the introduction when discussing lifestyle adjustments to underline whether we have strong or weak evidence about their effectiveness.

Response 4: Thank you for this helpful advice. 

In the introduction, we had already referred to the current evidence of the various lifestyle adjustments. We now adjusted or added statements on the evidence level in each area.

Lifestyle adjustments & progression, line 70 ff: “While the influence of lifestyle habits on MS progression in general still needs to be proven (8), low Vitamin D levels and smoking are the only factors with some weak to moderate evidence to be associated with progression (9). “

Nutrition, line 74 ff: “Similarly, systematic reviews on the effects of specific diets and supplements on disease risk and progression in pwMS have yielded conflicting results with no conclusive evidence (10, 11).”

Physical activity, line 81 ff: “Physical exercise has been investigated in detail, but most interventional studies included only small numbers of patients and investigated short-term effects. While improvement in health-related quality of life (QOL), fatigue, mood and numerous MS symptoms, such as stability of gait and prevention of falls, have been demonstrated in several systematic reviews (16–18), disease modifying or neuroprotective effects of physical exercise is still a matter of debate (8, 19) and the evidence is weak.”

Concerning the main theme “stress management” we added the following sequence: 

Stress management, line 96 ff: “However, systematic reviews yield to the conclusion that evidence of the influence of stress (management) on MS remains scarce due to major qualitative differences in methodology of the existing literature (29, 30). Taken together there is some weak to moderate evidence that especially major stressful life events trigger inflammatory activity in MS.”

Comment 5: Methods: Were there any changes in the interview schedule after the first five pilot interviews?

Response 5: Thank you for this question. No changes were made after the first five pilot interviews and these interviews could be fully included in the study sample. We clarified this statement in the manuscript:

Lines 155 ff: “The five interviews were fully included in the study sample and without further adjustment to the interview schedule.”

Comment 6: Methods: The topic guide does not align with the aims of the study. Only one question (4. Apart from DMTs, there are other therapies that can be used. These include alternative therapies or measures that change lifestyle habits.) discusses the lifestyle adjustments partly.

Response 6: Thank you for your point. The interview guideline was developed for the whole PexMS project which covered numerous aspects and not only for lifestyle. This was developed with the help of an expert panel on qualitative research of the University of Hamburg and an advisory board with neurologists, health researchers and pwMS. Open questions were designed following the Problem-centered interview (PCI) after Witzel et al. (2012), addressing the research themes of the whole project (experiences with MS diagnosis, daily life, and different treatment approaches). According to Witzel et al. (2012), interview guides assist the interviewer as a background framework for comparability and a tool for memorizing research themes throughout the interview. In the guides, the principle of openness should be realized by using broad formulations of introduction questions (such as “Apart from DMTs, there are other therapies that can be used. These include alternative therapies or measures that change lifestyle habits”), stimulating the interviewee to respond with his or her own words, interpretation, and leading the interview to his/her individual experiences and focus on these topics. Throughout the interview, the interviewer has the opportunity to refer to the thematic aspects of the patient’s narrative sequence by individual follow-up questions. According to Witzel et al. (2012), this approach stimulates the patient’s memory, creates coherence in his or her answers and establishes concrete reference between personal experiences and actions. Following the approach of PCI, we therefore developed a guide with broad introduction questions as a background reference for our interviewer. Throughout the interview, our experienced interviewer elaborated on the patients' individual narratives by follow-up questions. Following the PCI approach, we believe that the interviews gave a rich and nuanced insight of patients’ experiences with lifestyle adjustments and, thus, align the aims of this study. 

Comment 7: Methods: In the limitations, it is mentioned that this research project is part of a larger research project. Most transparency is needed in the methods section about the aims of the bigger project, how this study fitted within the larger study and how data were extracted.

Response 7: To avoid making the manuscript longer, we have referred to the study protocol [39], which describes the aim and all the steps of the PExMS project. However, we agree that the overall aim of the PExMS project should be more clarified in the method section and have therefore amended the paragraph as outlined in response 2 to reviewer 1. 

Comment 8: Methods: Again in the limitations, it is mentioned that the interviews were audio and video recorded, but this is not reported in the methods section.

Response 8: We appreciate the reviewer’s comment and carefully went through the method section again. In line 162 we had mentioned the conversion of video and audio tapes into verbatim transcripts. However, we agree, that this should be clarified earlier in the method section. Therefore, we have added “audio and video recorded” in line 147. 

Comment 9: Methods: The reporting of the study did not follow COREQ guidelines. Mainly there are missing criteria from Domain 1: 'Research team and Reflexivity'.

Response 9: We have added the missing COREQ criteria from line 156 ff:

“All interviews were conducted by a single interviewer (A.S.) – a female health scientist with expertise in qualitative research methods, who had no relationship to the patients in our study.”

Comment 10: Results: It is mentioned that each theme has 3 sub-themes, but I can’t see the third subtheme, ‘practiced lifestyle habits’, in any of the themes.

Response 10: We appreciate your concern. There were three main themes “Stress Management”, “Exercise and physical activity” and “Nutrition and supplements”. Each of these had three sub-themes “Starting new habits”, “Maintaining previous habits” and “Practiced lifestyle habits”. Starting new habits is defined as actively adjusting lifestyle habits after the diagnosis with MS, whereas maintaining previous habits is regarded as continuing lifestyle habits that had already been integrated in daily life prior to the diagnosis. These two sub-themes are described in detail in the results section. As also outlined in response 4 for reviewer 1 the third sub-theme “practiced lifestyle habits” is illustrated in the Figures 1-3 for each main theme. 

Comment 11: Results: The results section is very interesting and informative. However, I wonder whether re-arranging the theme will enable more of the details to come out. Instead of looking at specific lifestyle areas (stress, nutrition, exercise), the themes could be formed around the key elements underpinning all of these lifestyle choices, e.g. control, empowerment, information from health care professionals, benefits of not changing. Focusing on these themes can bring out more useful information on lifestyle adjustment that can be taken forward in future interventions and clinical practice.

Response 11: Thank you for giving your opinion about our result section. We have already thoroughly considered this option and would like to explain the reason for deciding to stay with the current structure. With this first paper of our study about lifestyle adjustments for pwMS, we would like to give an overview and explain in more detail the experiences of patients with MS with different lifestyle habits. Therefore, it was important for us to give an overview of the different lifestyle measures mentioned, reflecting also the influencing factors for decision making and their impact on everyday life. Thus, influences on everyday life might be very different for stress management strategies than for physical activity or nutrition. Also, the motives for or against the decision for a lifestyle measure differ within the main themes. We believe that the current structuring of the results therefore adds value for the reader: When looking at the results, the reader can understand exactly which factors influence the decision for each lifestyle category and what impact these habits had on the daily life of the patients in our study. This underlines the need to address the individual needs of patients for specific lifestyle measures and could further help the patient, but also the health care givers, to decide for or against specific lifestyle measures within a multimodal therapy.

Comment 12: Results: The title of the theme stress management is not accurate as the theme mainly discusses ways people use to manage their mental health more broadly and their relationships with others.

Response 12: We agree that the understanding of stress management techniques varies widely and that there is no common definition. In our study we mainly tried to refer to current literature, in which stress management techniques include techniques for mental health improvement. For example, a meta-analysis of stress management techniques for pwMS included the following techniques: “challenging and confronting negative or pessimistic thoughts, redefining life goals post MS diagnosis, and mindfulness acceptance (e.g. mindfulness-based stress reduction; MBSR) to train self-awareness and maximize coping ability. Behavioral techniques to relieve stress included progressive muscle relaxation and controlled breathing.” (Taylor et al., 2019, https://doi.org/10.1177/1359105319860185). Another meta-analysis regarding MS and stress management included the following techniques to their analysis: “Stress-management interventions were defined as psychological techniques employed to support MS patients in reducing emotional or physiological responses resulting from stressful events (including stress related to the disease process).” (Alison et al., 2014; https://doi.org/10.7224/1537-2073.2013-034). Therefore, we believe our choice of stress management instead of mental health is feasible.

Comment 13: Results: ‘Maintaining habits’ sub-theme is not discussed within the stress management theme.

Response 13: Thank you very much for the careful reading. The reason why “maintaining habits” is not discussed in the results is simply that our patients did not mention having used stress management strategies prior to their MS diagnosis. No patient did report having applied stress management techniques prior to MS.

Comment 14: Discussion: The discussion mainly repeats the findings. It would be helpful to discuss further theories and evidence around habit formation and behaviour change and how what we know about lifestyle adjustment in general fits (or not) within the MS population.

Response 14: We appreciate this constructive point. In the discussion we tried to link our findings to current literature and evidence with a couple of citations and thereby derive implications for practical use and potential limitations. Therefore, we do not agree that our discussion mainly repeats the findings. We agree that theories for habit formation and behavior change might be helpful to analyse patient experiences. We already referred to empowerment concepts in the discussion. In addition, we now integrated the HAPA model for behavior change as a possible approach to structure reported moderating factors for lifestyle behavior:

Lines 394 ff: “One theory analyzing behavior change for people with chronic diseases is the Health Action Process Approach (HAPA). In this concept, two different phases are postulated: First, the motivation phase, in which people develop their intentions to change the behavior, and second, the volitional phase, in which the intentions must be translated into action. Different factors contribute to intention formation in the motivation phase: Perceived task self-efficacy, risk perception and outcome expectations all of which can be influenced by acquiring new information. In the volitional phase, action planning and control, coping with barriers, social support and maintenance/recovery self-efficacy are correlated with successful behavior change (41). Recent research indicates that HAPA might be a useful framework for engaging pwMS to physical activity (42).”

Lines 407 ff: “According to HAPA model, knowledge expansion might help to define the expected outcome and risk factors to start a specific lifestyle habit.”

Lines 417 ff: “Perceived self-efficacy, a predictor for motivation in health behavior change (41), is further associated with the perception of improved physical and psychological health status (46) thereby also affecting QOL (47).”

Lines 435 ff: “Social support is correlated with successful behavior change according to the HAPA concept (43) and was another concise motivator for lifestyle adjustments in our study.”

---

## [Decision Letter · Decision Letter 1]

31 Mar 2022

PONE-D-21-10862R1Experiences of Persons with Multiple Sclerosis with Lifestyle Adjustment - A Qualitative Interview StudyPLOS ONE

Dear Dr. Elkhalii-Wilhelm,

Thank you for submitting your manuscript to PLOS ONE. After careful consideration, we feel that it has merit but does not fully meet PLOS ONE’s publication criteria as it currently stands. Therefore, we invite you to submit a revised version of the manuscript that addresses the points raised during the review process.

See comments below.  Please submit your revised manuscript by 15 May 2022. If you will need more time than this to complete your revisions, please reply to this message or contact the journal office at plosone@plos.org. Please include the following items when submitting your revised manuscript:A rebuttal letter that responds to each point raised by the academic editor and reviewer(s). You should upload this letter as a separate file labeled 'Response to Reviewers'.A marked-up copy of your manuscript that highlights changes made to the original version. You should upload this as a separate file labeled 'Revised Manuscript with Track Changes'.An unmarked version of your revised paper without tracked changes. You should upload this as a separate file labeled 'Manuscript'.If applicable, we recommend that you deposit your laboratory protocols in protocols.io to enhance the reproducibility of your results. Protocols.io assigns your protocol its own identifier (DOI) so that it can be cited independently in the future. For instructions see: https://journals.plos.org/plosone/s/submission-guidelines#loc-laboratory-protocols. Additionally, PLOS ONE offers an option for publishing peer-reviewed Lab Protocol articles, which describe protocols hosted on protocols.io. Read more information on sharing protocols at https://plos.org/protocols?utm_medium=editorial-email&utm_source=authorletters&utm_campaign=protocols.

We look forward to receiving your revised manuscript.

Kind regards,

Andrew Soundy

Academic Editor

PLOS ONE

Journal Requirements:

Additional Editor Comments (if provided):

Please consider the items from either COREQ (Tong et al) or SRQR (o'brien et al) and in particular the method section and any missing items of information e.g., strategies to enhance quality. If I have missed this apologies.

Could you update the Braun and Clarke reference noting they have a website and recent articles and the name of the analysis has changed.

Reviewers' comments:

Reviewer's Responses to Questions

**Comments to the Author**

1. If the authors have adequately addressed your comments raised in a previous round of review and you feel that this manuscript is now acceptable for publication, you may indicate that here to bypass the “Comments to the Author” section, enter your conflict of interest statement in the “Confidential to Editor” section, and submit your "Accept" recommendation.

Reviewer #1: All comments have been addressed

2. Is the manuscript technically sound, and do the data support the conclusions?

Reviewer #1: Yes

3. Has the statistical analysis been performed appropriately and rigorously? 

Reviewer #1: N/A

4. Have the authors made all data underlying the findings in their manuscript fully available?

Reviewer #1: Yes

5. Is the manuscript presented in an intelligible fashion and written in standard English?

Reviewer #1: Yes

6. Review Comments to the Author

Reviewer #1: (No Response)

7. PLOS authors have the option to publish the peer review history of their article (what does this mean?). If published, this will include your full peer review and any attached files.

Reviewer #1: **Yes: **Silvia Poli

---

## [Author Response · Author response to Decision Letter 1]

29 Apr 2022

Subject: Edits requested on the submission PONE-D-21-10862R1

Dear Ms. Rose Ann Joyce Sagun Puetes, 

Thank you for giving us the opportunity to submit a revised draft of our manuscript entitled “Experiences of Persons with Multiple Sclerosis with Lifestyle Adjustment - A Qualitative Interview Study” to PLOS ONE.

We believe that the paper has significantly improved with your help. Please find below a point-by-point response and let us know if you have any further concerns or questions. 

Best wishes

Saskia Elkhalii-Wilhelm

Point by point response 

Editor’s comments:

Comment 1: Please consider the items from either COREQ (Tong et al) or SRQR (o'brien et al) and in particular the method section and any missing items of information e.g., strategies to enhance quality. If I have missed this apologies.

Response 1: We appreciate this advice and agree about the importance of referring to specific criteria for reporting qualitative data to enhance quality. Now we have clarified how we specifically referred to the COREQ checklist from Tong et al. (2007) in a newly created “S2 Appendix”.

Comment 2: Could you update the Braun and Clarke reference noting they have a website and recent articles and the name of the analysis has changed.

Response 2: We thank you for taking the time to evaluate our manuscript and for this valuable comment. Indeed, Braun and Clarke call their approach also ‘reflexive thematic analysis’ to show that this approach is quite theoretically flexible to address different types of research questions. Thematic analysis can be approached in an inductive, deductive, semantic, latent realist or in a constructionist way (Braun and Clarke, 2022). The coding and theme development in our article was directed by existing concepts, on the one hand, and by the content of the data, on the other hand. We added that this form of analysis is reflexive and updated the reference list by adding recent articles of Braun and Clarke (2019, 2021) and referring to their website www.thematicanalysis.net on p. 7, lines 157-158:

“Data were analyzed inductively and deductively according to the six-step (reflexive) thematic analysis of Braun and Clarke [43-46]”

---

## [Decision Letter · Decision Letter 2]

13 May 2022

Experiences of Persons with Multiple Sclerosis with Lifestyle Adjustment - A Qualitative Interview Study

PONE-D-21-10862R2

Dear Dr. Elkhalii-Wilhelm,

We’re pleased to inform you that your manuscript has been judged scientifically suitable for publication and will be formally accepted for publication once it meets all outstanding technical requirements.

Kind regards,

Andrew Soundy

Academic Editor

PLOS ONE

Additional Editor Comments (optional):

Reviewers' comments:

Reviewer's Responses to Questions

**Comments to the Author**

1. If the authors have adequately addressed your comments raised in a previous round of review and you feel that this manuscript is now acceptable for publication, you may indicate that here to bypass the “Comments to the Author” section, enter your conflict of interest statement in the “Confidential to Editor” section, and submit your "Accept" recommendation.

Reviewer #1: All comments have been addressed

2. Is the manuscript technically sound, and do the data support the conclusions?

Reviewer #1: Yes

3. Has the statistical analysis been performed appropriately and rigorously? 

Reviewer #1: N/A

4. Have the authors made all data underlying the findings in their manuscript fully available?

Reviewer #1: Yes

5. Is the manuscript presented in an intelligible fashion and written in standard English?

Reviewer #1: Yes

6. Review Comments to the Author

Reviewer #1: (No Response)

7. PLOS authors have the option to publish the peer review history of their article (what does this mean?). If published, this will include your full peer review and any attached files.

Reviewer #1: **Yes: **Silvia Poli

---

## [Editor Report · Acceptance letter]

18 May 2022

PONE-D-21-10862R2 

Experiences of Persons with Multiple Sclerosis with Lifestyle Adjustment – A Qualitative Interview Study

Dear Dr. Elkhalii-Wilhelm:

I'm pleased to inform you that your manuscript has been deemed suitable for publication in PLOS ONE. Congratulations! Your manuscript is now with our production department. 

Kind regards, 

on behalf of

Dr. Andrew Soundy 

Academic Editor

PLOS ONE